# Multi-scale wastewater surveillance at a Bangkok tertiary care hospital: A potential sentinel site for real-time COVID-19 surveillance at hospital and national levels

**Quinton Hayre**[1], **Supaporn Wacharapluesadee**[1], **Piyapha Hirunpatrawong**[1], **Ananporn Supataragul**[1], **Opass Putcharoen**[1,2], **Leilani Paitoonpong**[1,2]*

**1** Thai Red Cross Emerging Infectious Disease Clinical Center, King Chulalongkorn Memorial Hospital, Rama IV Road, Pathumwan, Bangkok, Thailand, **2** Division of Infectious Diseases, Department of Medicine, Faculty of Medicine, Chulalongkorn University, Rama IV Road, Pathumwan, Bangkok, Thailand

* leilani.p@chula.ac.th

## Abstract

Wastewater-based epidemiology is a valuable tool for population-level pathogen surveillance, complementing clinical methods. While most sampling focuses on municipal wastewater treatment plants, emerging evidence suggests wastewater collected from hospital settings can lead to targeted clinical interventions. To investigate wastewater pathogen surveillance in hospital settings further, we tracked the presence and concentration of SARS-CoV-2 RNA in wastewater across multi-scale sample sites within a large, public tertiary care hospital in Bangkok, Thailand. From July 2022 to May 2023, weekly wastewater samples (n=392) were collected from various sample sites including clinical and non-clinical facilities, as well as the hospital's wastewater treatment plant. Influent wastewater at the hospital's wastewater treatment center yielded the most consistent SARS-CoV-2 RNA detection across all sample sites, with detection in all 26 samples. Despite varied building usage patterns, significant moderate negative correlations were found in 90% (9/10) of sample sites between wastewater RT-PCR cycle threshold values and clinical case data from hospital and national reports. Targeting specific buildings yielded distinct data trends, indicating their potential to offer complementary insights into viral shedding and transmission among clinical and non-clinical sub-populations within a hospital campus. Our findings suggest that hospital wastewater-based epidemiology reflects broader community disease trends, which may be especially useful in regions with limited municipal wastewater treatment coverage. Large tertiary care hospitals could serve as effective and cost-efficient sentinel surveillance sites for future pathogen monitoring, guiding public health actions.

## Introduction

Wastewater pathogen surveillance has been used for several decades to monitor the presence of infectious disease transmission as an indicator of disease prevalence in the community

**Data availability statement:** All wastewater data generated during this study are included in this published article and its supplementary materials. Additional data analyzed in this current study are available in the WHO dashboard, https://data.who.int/dashboards/covid19/data. (26).

**Funding:** This work was supported by the Thai Red Cross Research Committee fiscal year 2022 (TRC2022 to LP, OP, SW, and PH), the World Health Organization (WHO Reference 2022/1278782 to OP and SW), and the US Centers for Disease Control and Prevention (Cooperative Agreement No. U01GH002402 to SW, OP). The funders had no role in study design, data collection and analysis, decision to publish, or preparation of the manuscript.

**Competing interests:** The authors have declared that no competing interests exist.

independent of symptomatic presentation or healthcare utilization [1]. The detection and quantification of viral and bacterial genetic material found in toilet water, or blackwater, has allowed for public health action, with a long history of successful poliovirus interventions [1,2].

During the Coronavirus Disease 2019 (COVID-19) pandemic, wastewater-based epidemiology (WBE) was quickly adapted to monitor severe acute respiratory syndrome coronavirus 2 (SARS-CoV-2) RNA at the population level [3,4]. Despite its primary classification as a respiratory-transmitted virus, the fecal shedding of SARS-CoV-2 among certain infected individuals has been well documented in clinical observations [5]. Causes of variability in the viral shedding schedule and quantity in feces among infected individuals remain uncertain, with hypothesized linkages to the variant of infection, vaccination status, disease severity, age, and presence of gastrointestinal symptoms [5–9]. Wastewater studies from municipal wastewater treatment plants (WWTPs) and other sewersheds with large contributing populations have successfully acted as an early warning signal for community outbreaks largely due to the capturing of viral loads from asymptomatic infected individuals [10,11].

Building-specific WBE has the potential to offer surveillance data representing specific populations allowing for tailored policy interventions [12–14]. However, correlating wastewater trends with community clinical data has shown occasional signs of difficulty in sewersheds with smaller population coverage due to greater susceptibility to isolated outbreaks and individual viral load variability [15,16].

Similarly, WBE from hospital campuses enables targeted surveillance of hospital patients, visitors, and healthcare workers. Compared to broader community settings, hospital clinical surveillance data is likely to be timelier and more inclusive of the sewershed population allowing for more informative correlation analyses on pathogen shedding behavior detected in wastewater [15,17–19]. Preliminary WBE studies have detected higher than expected levels of SARS-CoV-2 RNA in wastewater from healthcare settings, highlighting the risk for nosocomial outbreaks with potentially dangerous impacts on high-risk and vulnerable patient populations [17,18,20,21]. Such outbreaks could limit other necessary hospital procedures [22]. Existing studies in Thailand that explore the relationship between hospital wastewater trends and clinical data are limited and have predominantly focused on bacterial pathogens [23,24]. This underscores the necessity for further research, particularly within public tertiary care hospitals, to evaluate their potential as sentinel surveillance sites for disease prevalence. Our study addresses this gap by developing a novel building-specific hospital wastewater surveillance program in Thailand. It aims to improve understanding of viral load contributions from both patient and non-patient populations across the hospital campus. By providing policymakers with precise and comprehensive data, this initiative facilitates informed interpretations of future wastewater findings, thereby enhancing public health surveillance and response strategies.

Our study aims to advance the understanding of hospital WBE's suitability for future investigation by examining clinical and non-clinical hospital buildings and building clusters at our selected hospital, a large and sprawling public tertiary care hospital campus in Bangkok, Thailand. Specifically, we aim to evaluate several sample sites within our selected hospital to provide valuable insights into (1) SARS-CoV-2 RNA detection and concentration variability and (2) correlation with hospital and national-level COVID-19 clinical trends. We anticipate our findings will offer valuable insights into building-specific wastewater patterns within tertiary hospital settings, paving the way for future targeted investigations into tracking human pathogens detectable in wastewater.

## Materials and methods

### Study setting

The hospital examined in this study is a public tertiary care and academic hospital in central Bangkok, Thailand. The hospital serves approximately 1.5 million outpatients annually and holds over 1,500 inpatient beds and 400 outpatient examination rooms. In 2022 and 2023, inpatient bed occupancy rates in the hospital averaged 70.61% and 66.48%, respectively, nearing or remaining below the hospital's target rate of 70% for efficient operations. The hospital campus encompasses 43 buildings dedicated to patient care, teaching, dormitories, public cafeterias, and support services. The hospital was chosen for this study because it serves a large and economically diverse patient population, is centrally located in a densely populated metropolitan area, and has a distinct segmentation of buildings by clinical and non-clinical functions. This made it an ideal location to examine wastewater trends in alignment with clinical trends across different community and patient populations.

Ten wastewater sample sites were chosen across the hospital campus (Fig 1). Sample sites included various configurations of building wastewater systems: six standalone buildings, each with independent wastewater tanks (designated as B1 to B6); two clusters, each consisting of three buildings sharing a common wastewater tank (designated as C1 and C2); and both influent (TCi) and effluent (TCe) samples from the hospital's wastewater center. Table 1 provides details on the use types of each building, offering insights into the patient and non-patient populations that contribute to the wastewater collections at each sample site. Samples from the TCe sample site consisted of post-chlorination effluent wastewater, while the remaining nine sample sites were raw influent wastewater.

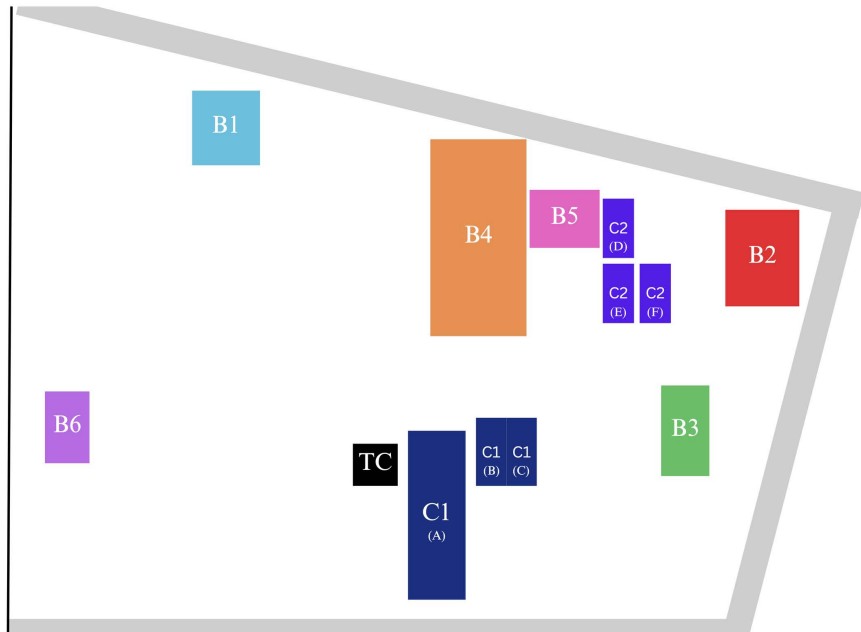

**Fig 1. Map of Hospital Campus Buildings Included in the Wastewater Study, Bangkok, Thailand.** Individual buildings sampled are marked with a "B." For buildings sharing a sample site, or building clusters, labels include a "C" followed by a unique letter in parentheses specific to each building in the cluster. The wastewater treatment center (TC) represents combined wastewater from all buildings and building clusters on the hospital campus.

**Table 1. Hospital sample site description.**

| Sample Site | Specialties | Gross Floor Area ($m^2$) | COVID-19 Ward |
|---|---|---|---|
| B1 | Geriatric (primary use); pharmacy; public food court; | 40,110 | No |
| B2 | Outpatient departments | 51,780 | No |
| B3 | COVID-19 unit (outpatient, intensive care[a], inpatient[b], & pharmacy); research laboratory | 5,570 | Yes |
| B4 | Inpatient departments (including COVID-19 emergency, intensive care[a], & inpatient[b]); conference center; public food court; pharmacy | 239,560 | Yes |
| C1 | Building A: Public food court (primary use); medical equipment storage | 38,925 | No |
|  | Buildings B and C (connected): Cancer outpatient and testing | 11,605 |  |
| TCe | Wastewater treatment center (post-treatment effluent) | N/A | N/A |
| TCi | Wastewater treatment center (pre-treatment influent) | N/A | N/A |
| B5 | Pediatric, primarily oncology and complex heart conditions; pediatric COVID-19 outpatient | 47,285 | Yes |
| B6 | Physician-in-training dormitory; optional quarantine for COVID-19-positive staff | 34,100 | Yes |
| C2 | Buildings D and E: Non-patient office use | 2,230 + 1,370 | No |
|  | Building F: Liver disease clinical and research center | 340 |  |

[a]Moved from B3 to B4 in December 2022.

[b]Moved from B3 to B4 in February 2023.

The hospital wastewater treatment center functioned properly throughout the study. A vacuum sewer system was used to route wastewater from building-specific or building cluster-specific holding tanks to the hospital wastewater treatment center at least once a day. During the study period, wastewater inflow at the hospital wastewater treatment center, 2876 to 3558 $m^3$/day, was consistent and below the maximum wastewater processing capacity, 5600 $m^3$/day. Wastewater was treated through an activated sludge process coupled with chlorination. Influent wastewater from the hospital wastewater treatment center was collected in a 3000 $m^3$ wastewater collection pond before treatment. Non-blackwater wastewater, referred to as greywater, was estimated to be a major contributor to all sample sites. Stormwater did not contribute to the wastewater collected in this study.

Clinical services for COVID-19 treatment in the selected hospital underwent significant changes during the study. The COVID-19 intensive care unit (ICU) and inpatient department were moved from building B3 to B4 on 01 December 2022 and 01 February 2023, respectively. This shift reduced the number of COVID-19 inpatient beds from 16 to 10 reflecting an overall relaxation of COVID-19 clinical policy due to a reduction in cases compared to the peak of the pandemic. Throughout the study period, patients were required to test for COVID-19 before being admitted to building B3 or B4. COVID-19 patients were admitted either to a single-occupancy room or to a shared room with other COVID-19 patients, referred to as a cohort ward. Patients at a higher risk for generating aerosol were admitted to a single-occupancy airborne isolation unit.

## Wastewater sample collection

Samples were collected weekly from 07 July 2022 to 30 May 2023. For each sample site, 250 mL grab samples were collected with a stainless-steel sampling dipper between 09:30 AM and 10:30 AM. Samples were collected in the morning to effectively capture human contributions to the wastewater systems, aligning with facility opening times and typical daily human

activity patterns. Sampling took place weekly on Tuesdays with the exception of the initial collection date. Wastewater samples were transferred to high-density polyethylene plastic containers. Samples were stored and transported using a triple-layer packaging method under strictly controlled conditions to maintain integrity. During transport, the samples were kept at 4°C until processing. No preservatives or stabilizing agents were used. All samples were processed within 24 hours of collection, ensuring that their stability and integrity were preserved until analysis.

Sample sites were phased into the study in two stages. Stage 1 comprised of six sample sites included for weekly sampling across all 48 collection dates from 07 July 2022 to 30 May 2023, namely B1-B4, C1, and TCe. Stage 2 sample sites, B5, B6, C2, and TCi, were included for 26 sample collection dates from 06 December 2022 to 30 May 2023.

## Wastewater RNA extraction

Wastewater RNA isolation was performed using the ZR Urine RNA Isolation Kit (Zymo Research, Orange, CA, USA), following the manufacturer's protocol [25]. Wastewater samples were thoroughly mixed by inversion and briefly spun at 3000g for 10 minutes to remove the precipitate. The supernatant was further used for nucleic acid extraction processing. Prior to the extraction process, 5 μL of lentivirus reference RNA was added to each sample as an internal control. To isolate the RNA, 30 mL of the clean wastewater sample was filtered through a 1.6 μm ZRC-GF Filter. Then, 700 μm of Urine RNA Buffer was pushed through the filter. The flow-through was collected in a nuclease-free tube and 700 μm of ethanol was added. The RNA was purified using the Zymo-Spin IC Column undergoing sequential washes using the RNA Prep Buffer and RNA Wash Buffer. RNA was eluted with 50 μm of DNase/RNase-Free Water. Extracted RNA was immediately tested for SARS-CoV-2 detection. The remaining RNA was kept at -80°C for further analysis.

## Real-time RT-PCR SARS-CoV-2 detection

The Fosun COVID-19 RT-PCR Detection Kit (Fosun, Shanghai, China) was used for real-time reverse-transcription polymerase chain reaction (RT-PCR) detection of the ORF1ab, E, and N gene in SARS-CoV-2 RNA. The manufacturer's protocol was followed. Briefly, 10 μL of the extracted RNA was combined with 14 μL of the SARS-CoV-2 reaction reagent and 6 μL of the RT-PCR enzyme mixture. The Bio-Rad CFX 96 PCR instrument (Bio-Rad, Hercules, CA, USA) was used for PCR amplification and detection. Thermal cycling conditions included reverse transcription at 50°C for 15 minutes, followed by an initial denaturation at 95°C for 3 minutes. This was followed by 5 cycles of 95°C for 5 seconds and 60°C for 40 seconds, then 40 cycles of 95°C for 5 seconds and 60°C for 40 seconds. A cycle threshold (Ct) value $\leq$ 36 was the provided threshold to detect COVID-19. SARS-CoV-2 was determined to be detected in a sample when at least two of the three included genes were within the accepted threshold.

Criteria for quality control procedures were followed as indicated by the manufacturer. Each test run included a positive control for SARS-CoV-2, which required a Ct value of 32 or less for the detection of all three SARS-CoV-2 genes, and a negative control, which was valid only when showing no amplification in any of these genes. An internal control was integrated into each sample to evaluate RNA extraction efficiency and ensure that no PCR inhibitors were present during amplification. The internal control was deemed valid with a Ct value of 32 or less. If any of these criteria were not met, the test was considered invalid, prompting a review of the experimental conditions and a repetition of the test.

### Clinical COVID-19 case data sources

Weekly laboratory-confirmed COVID-19 cases in Thailand from the study period were retrieved on 20 May 2024 from the WHO, based on data from the Ministry of Public Health of Thailand [26]. Laboratory-confirmed COVID-19 patient cases within the studied hospital were also obtained from twice-monthly reports to the Ministry of Public Health of Thailand. COVID-19 cases from the selected hospital accounted for 4.89% of laboratory-confirmed cases nationally (11140/227803) over the study period. Directly employed healthcare workers were the only non-patient population present on the hospital campus where COVID-19 case data was recorded. Therefore, weekly COVID-19 cases among directly employed healthcare workers at the selected hospital were obtained based on the self-report and five-day quarantine protocol. Clinical data used in this study were anonymous and approved by the Institutional Review Board of the Faculty of Medicine, Chulalongkorn University (IRB. No.221/65).

### Data analysis

Spearman's rank correlation was employed to assess the relationships between clinical COVID-19 case data from patients and personnel at the selected hospital, as well as national reports, with wastewater N gene Ct values. Each sample site was considered independently. A Bonferroni correction was applied to account for the multiple comparisons at each site. The Fisher Z-transformation was used to calculate Bonferroni-adjusted 95% confidence intervals for the Spearman correlation coefficients [27]. Wastewater samples where the N gene was not detected were assigned a Ct value of 37, above the cut-off Ct value of 36, for correlation analyses.

A comparison of SARS-CoV-2 lineage identification among confirmed COVID-19 clinical specimens was included to assess the potential variability in lineage trends between the selected hospital and the greater community. Whole genome data was retrieved on 01 May 2024 from the selected hospital's SARS-CoV-2 genomic surveillance data submitted to the NCBI GenBank and GISAID databases. COVID-19 GISAID data was retrieved on 23 May 2024 and included all specimens collected from July 2022 through May 2023 in Thailand for which lineages were assigned [28]. Lineages were summarized according to Nextstrain clade groupings using Pango lineage nomenclature [29,30].

All data analysis was conducted using the R statistical software (version 4.2.3). Data was cleaned, analyzed, and plotted using the 'dplyr', 'ggplot2', and 'stats' packages.

## Results

### Clinical COVID-19 trends from the selected hospital and the greater community

COVID-19 case reports from the selected hospital and Thailand were acquired to assess clinical data at various scales. The complete datasets are available in the S1 Table, S2 Table and S3 Table. Similar trends were found between COVID-19 clinical case counts, among patients and healthcare workers, from the selected hospital and Thailand during the study period (Fig 2). For each of the three clinical datasets, COVID-19 cases declined from July to October 2022, followed by a slight increase that peaked in December 2022. From February through March 2023 cases were consistently low followed by a steady increase for the remainder of the study period.

Genomic surveillance in the selected hospital conducted lineage identification on 205 clinical respiratory specimens from confirmed COVID-19 patients during the study period using whole genome sequencing. Accession numbers are available in the S4 Table. Genomic surveillance was

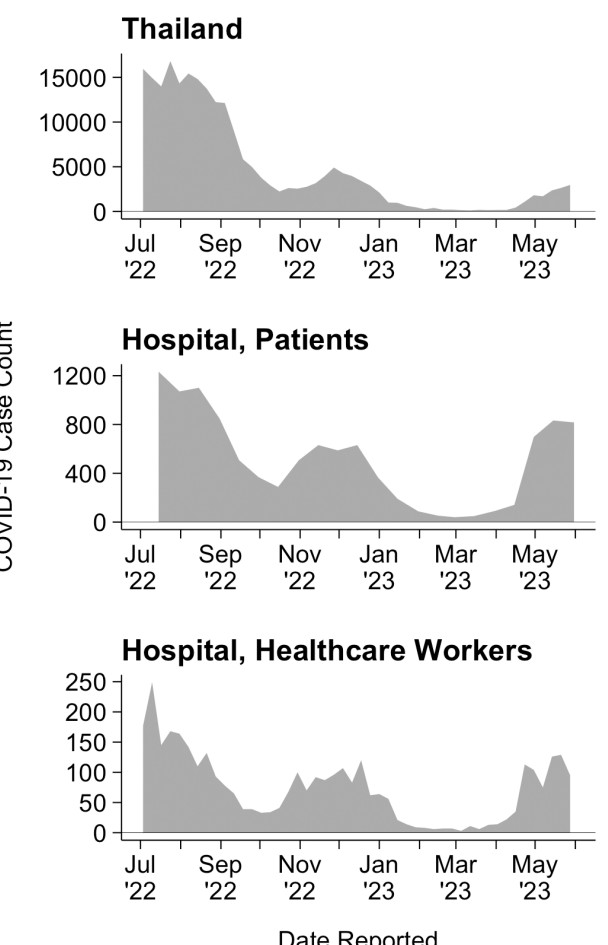

**Fig 2. COVID-19 Clinical Case Reports from July 2022 through May 2023.** COVID-19 case reports from the hospital's patients were accessed from twice monthly mandatory reporting to the Ministry of Public Health of Thailand. Healthcare worker cases from the hospital were voluntarily disclosed and reported weekly. National cases in Thailand were accessed from the WHO database based on weekly reports.

reduced during periods of low COVID-19 patient counts in the hospital, making comparisons between the datasets less informative. A general alignment between genomic surveillance from the selected hospital and GISAID entries from Bangkok and, more broadly, Thailand suggests that the selected hospital aligns well with the prevalent genomic trends in the community (Fig 3).

## SARS-CoV-2 RNA detection and quantification in wastewater samples

A total of 392 wastewater samples were collected from ten sample sites representing six independent buildings, two building clusters, and an influent and effluent site from the wastewater treatment center at the selected hospital. All samples met the established quality control criteria, confirming the validity of the test results. SARS-CoV-2 RNA was detected by RT-PCR in 85.7% (336/392) of the total samples. Raw RT-PCR wastewater data is made available in the S5 Table.

Ct values for each gene detected in SARS-CoV-2-positive wastewater samples were measured using RT-PCR. A median Ct value of 29.32 (IQR: 27.31 – 31.06, n = 318), 29.48 (IQR: 27.51 – 31.04, n = 318), and 27.57 (IQR: 25.38 – 29.44, n = 332) was measured for the ORF1ab, E, and N genes where SARS-CoV-2 RNA was detected, respectively. The N gene yielded lower

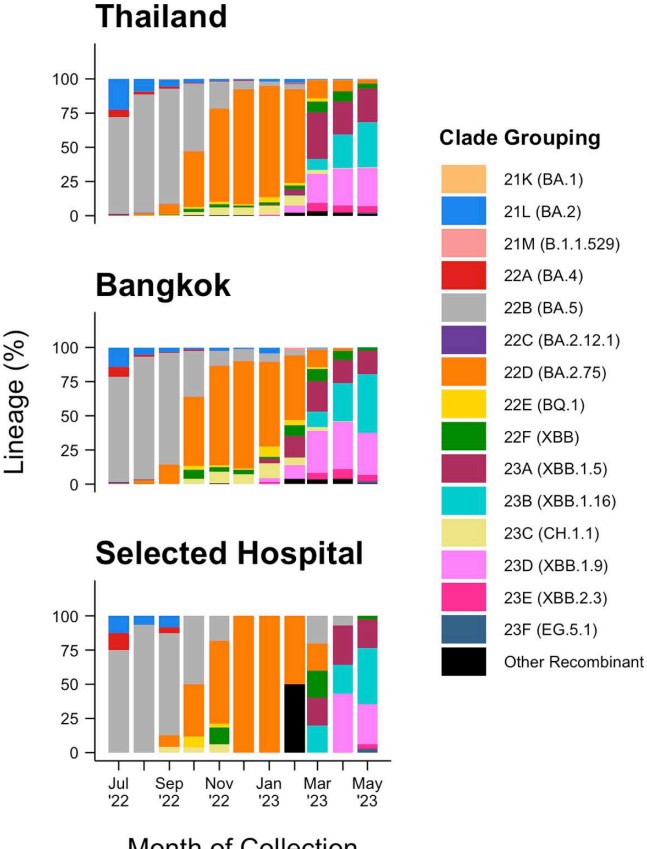

**Fig 3. Genomic Comparison of SARS-CoV-2 Clinical Lineages Between the National, Metropolitan, and Hospital Levels. Sequenced COVID-19 clinical specimens collected in Thailand and Bangkok were retrieved from GISAID entries. Genomic surveillance data from COVID-19 patients at the selected hospital were retrieved from the GISAID and NCBI GenBank databases.**

median Ct values and higher detection rates among positive samples than the ORF1ab and E genes resulting in the N gene's selection as the target gene used for analysis. N gene Ct value outlier values were retained for analysis due to expected natural variation and shifts in pathogen prevalence (Fig 4).

Median N gene Ct values for each sample site are described (Fig 4). Sample site B3, the hospital's dedicated COVID-19 clinic, had the lowest median N gene Ct value of 25.02. This was followed by sample sites C2 and then B6, both of which primarily served hospital healthcare workers. At the hospital level, the median N gene Ct value of influent wastewater, TCi, was lower than that of effluent, TCe, by a value of 3.43 (26.70 and 30.13, respectively).

Temporal variations in SARS-CoV-2 RNA detection and N gene Ct values are depicted in a heatmap (Fig 5). An overall reduction in SARS-CoV-2 RNA detection from mid-January to mid-April 2023 was found. This trend is consistent with low levels of COVID-19 reported in both hospital and national case data (Fig 2). During this period, occasional spikes in SARCoV-2 concentration, indicated by low N gene Ct values, were found in wastewater samples from sample sites B4 and B6.

At the hospital campus level, sample site TCi detected SARS-CoV-2 RNA in 100% (26/26) of its samples since its introduction to the study in Stage 2. Consistent detection of

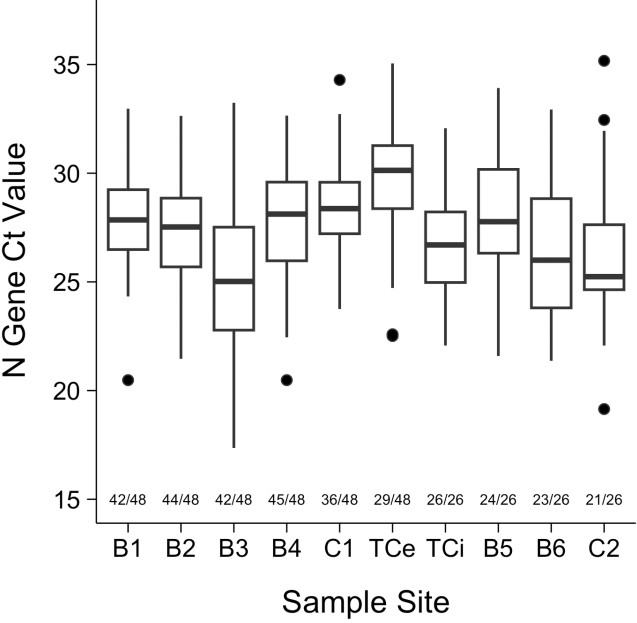

**Fig 4. Box Plot of N Gene Ct Values in Wastewater, by Sample Site.** The labeled proportions represent SARS-CoV-2 detection rates for each sample site.

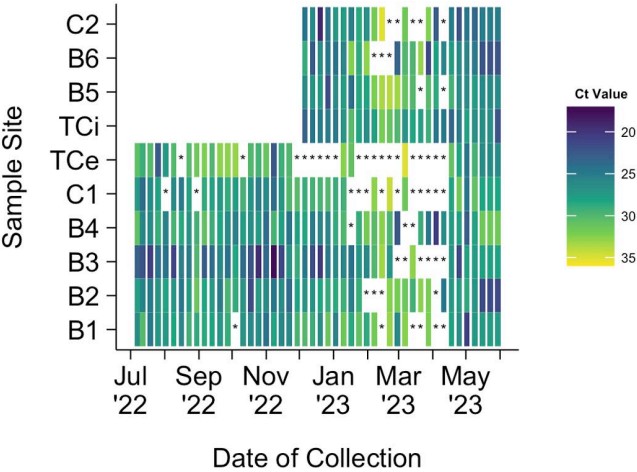

**Fig 5. Heat Map of N Gene Ct Values from Wastewater Samples Collected at Various Sample Sites in the Selected Hospital.** An asterisk indicates that SARS-CoV-2 RNA was not detected.

SARS-CoV-2 RNA was maintained despite low clinical COVID-19-positive patient case reports in February and March 2023. Effluent wastewater from the hospital wastewater treatment center, sample site TCe, yielded a total SARS-CoV-2 RNA detection rate of 62.5% (30/48) and a Stage 2 detection rate of 42.3% (11/26). Therefore, the hospital wastewater treatment center successfully reduced SARS-CoV-2 genetic material to undetectable levels in the effluent wastewater for 15 out of 26 samples, where the genetic material was initially detected in the influent wastewater in Stage 2.

## Correlations between SARS-CoV-2 RNA in wastewater and hospital-wide and nationwide COVID-19 reports

The comparison of wastewater sample N gene Ct values at each hospital sample site was conducted to assess any correlation with the clinical trends observed within the selected hospital and across Thailand. This analysis aimed to investigate whether there is a monotonic association between viral load detected in hospital wastewater and the incidence of cases within both the specific clinical setting and the wider national community.

To further explore this relationship, Spearman's rho correlation coefficient was computed to evaluate the correlation between the N gene Ct values at each sample site and the specified clinical variables. The measured Spearman's rho correlation coefficients for each tested relationship, regardless of significance, demonstrated a negative correlation (Fig 6). The details of the correlation analysis for each sample site are shown in the S6 Table.

All sample sites, except B4, yielded significant and moderate ($0.4 \leq$ |rho| $< 0.8$) correlations with clinical reporting within the hospital and in Thailand (Fig 6). The strongest correlations among the three relationships tested were split between sample sites B5 and C2. The COVID-19 weekly case report from Thailand was most strongly associated with N gene Ct value variations from sample site B5, the hospital's pediatric clinic, rho(24) = -0.77, p < 0.001. The hospital's patient and healthcare worker cases were most strongly associated with N gene Ct values from building cluster C2, rho(24) = -0.74, p < 0.001 and rho(24) = -0.73, p < 0.001, respectively.

Notably, sample site B4, the primary inpatient building in the hospital, yielded weak (|rho| < 0.4) and non-significant correlations with COVID-19 case trends from all three datasets. Wastewater samples from B4 had a median N gene Ct value of 28.12, similar to the study median (27.57), and a high SARS-CoV-2 detection rate of 94%.

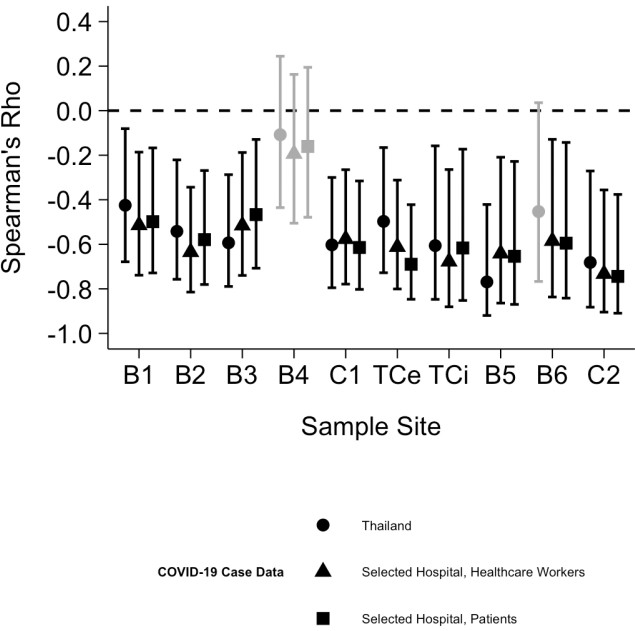

**Fig 6. Spearman's Rho Coefficient for the Correlation Between Wastewater Sample SARS-CoV-2 N Gene Ct Values at Various Sample Sites and COVID-19 Clinical Reporting at a Hospital and National Level.** Significant correlations after Bonferroni correction (p < 0.017) are indicated by black shapes. Bonferroni-adjusted 95% confidence intervals for the Spearman correlation coefficients were calculated using the Fisher Z-transformation.

## Discussion

This study assessed the feasibility of using wastewater pathogen surveillance within a tertiary care hospital among a variety of building use types. Ten sample sites were selected to capture diverse populations and wastewater matrices, while also considering budget constraints and hospital permissions. The diverse and specific building functions across the sprawling hospital campus provided unique opportunities for detailed wastewater investigation. Primarily, our analysis measured viral concentration spikes in wastewater at each sample site. We evaluated how these spikes correlate with available clinical data across different spatial resolutions to determine the influencing factors. SARS-CoV-2 was selected as the viral pathogen for this investigation due to its clinically confirmed prevalence in the community and ongoing clinical surveillance within the hospital.

We found varying SARS-CoV-2 RNA detection rates and concentrations, as approximated by measured N gene Ct values, at the hospital campus level. SARS-CoV-2 RNA was detected in each influent sample from the hospital wastewater treatment center, despite COVID-19 patient reports in the selected hospital dropping below 4% of their July 2022 peak levels in February and early March 2023. Given the high SARS-CoV-2 RNA detection rate, sample collection from influent wastewater at the selected hospital's wastewater treatment center is recommend for ongoing monitoring of SARS-CoV-2 and other pathogens at the hospital campus level. Treatment via activated sludge and chlorination at the hospital wastewater treatment center was not consistently effective in reducing SARS-CoV-2 RNA to undetectable levels. Although there is no conclusive evidence to date that SARS-CoV-2 has been transmitted via wastewater, further investigation may be warranted [31]. Evaluating the success of the hospital wastewater treatment center in eliminating other human pathogens, such as certain pathogenic bacteria with higher infection risks from wastewater contamination, remains essential [32].

At the building and building cluster levels, viral concentration variation was found between sample sites as indicated by median N gene Ct values among positive samples. Variability in the data may be attributed to several factors. One major factor includes sample dilution resulting from greywater or blackwater contributions from uninfected individuals [15]. Seasonal changes in population size and behavior among subgroups of patients and hospital personnel, along with the impacts of shifting weather conditions on wastewater holding tanks, may also contribute to this variability [33,34]. Furthermore, differences in viral loads captured in wastewater may significantly influence the results. This is supported by clinical findings linking SARS-CoV-2 viral shedding to factors such as disease severity, immune status, and stage of infection [8].

Significant correlations were measured between wastewater N gene Ct values and clinical case reports among various population groups and scales (Fig 6). To limit previously stated concerns related to inter-sample site wastewater variations, all correlation analyses were calculated relative to each sample site [35].

Wastewater from sample site B5, the pediatric care building, yielded the strongest significant correlation with clinical reporting from Thailand indicating consistency with trends in the broader community. Non-critical pediatric inpatients positive for COVID-19 were moved to building B4 according to hospital policy, while those in critical condition were placed in the pediatric ICU in B5. Patients in building B5 primarily included complicated inpatient pediatric cases of cancer and heart conditions. Studies on COVID-19 have pointed to potentially higher detection rates of SARS-CoV-2 in fecal samples coupled with higher probabilities of possessing gastrointestinal symptoms in children than in adults [6,9]. Therefore, children may make fecal contributions to the hospital sewage system while seeking treatment at higher rates and with greater detectability than adults allowing for a more complete capture of present

COVID-19 cases within a building by hospital wastewater surveillance. Additionally, hospital policy required one guardian to accompany each pediatric inpatient, with these guardians being screened for COVID-19 until 25 October 2022. From that date onward, additional visitors were permitted to B5. The viral contributions of these visitors and pediatric patients to the sewage system may partially explain the strong observed correlation between wastewater data and clinical trends in the greater community.

Sample site C2, a cluster of buildings primarily related to hospital administrative use, yielded the strongest significant correlations for the remaining two tested relationships, namely with cases from the hospital's patients and healthcare workers. A previous investigation indicated that COVID-19 infections with milder symptoms may have a higher likelihood of positive SARS-CoV-2 RNA signals in their stool [36]. This may partially explain the correlation between infected and non-isolating hospital healthcare worker case numbers. The strong association with patient case reports may be a result of the close mirroring found between patient and healthcare worker cases (Fig 2). Further investigation may be warranted into the origin of infection for hospital healthcare workers.

Notably, B4 was the only sample site not associated with a statistically significant correlation to any of the relationships tested. Sample site B4 primarily serves as the hospital's main inpatient department with over 1000 inpatient beds, 10 of which are dedicated towards inpatients admitted with or who have acquired COVID-19 as of February 2023. B4 is the largest building on the hospital's campus, 239,560 square meters of gross floor area, and contains a public food court, conference center, and the hospital's emergency department. Clinical data on COVID-19 nosocomial outbreaks among inpatients was not consistently recorded. Previous investigations have employed wastewater surveillance to detect nosocomial outbreaks of COVID-19 [17,18]. While insufficient data did not permit further investigation, the lack of direct correlation between spikes in wastewater viral concentrations in the inpatient department and clinical trends in the hospital and broader community could suggest isolated, building-specific transmission. This is particularly plausible given the large inpatient population and the regular movement of community members and hospital personnel within this single building.

Alternatively, hospital personnel employed through a third party, mainly cleaning and repair staff, are not included in the hospital's reporting of healthcare worker cases. Therefore, an estimated 600 unique personnel who work in B4 each week may contribute to wastewater viral loads without being captured by the hospital's clinical reports.

The varying coverage of wastewater surveillance programs between low- and middle-income countries (LMICs) and high-income countries highlights the need for innovative WBE strategies. In Bangkok, municipal WWTPs had an estimated population coverage of 49.6% [37]. The population coverage of wastewater surveillance programs is often lower in LMICs compared to high-income countries, where coverage can reach up to 95% in some metropolitan regions [38, 39]. A previous study of Bangkok's 19 municipal WWTPs from January to April 2021 found the strongest correlation between wastewater SARS-CoV-2 viral loads and COVID-19 trends in the community (Spearman's rho = 0.64) after a 21-day lag period [37]. Our study found stronger correlations between N gene Ct values and COVID-19 trends in the community at two building-level sample sites, B5 (Spearman's rho = -0.77) and C2 (Spearman's rho = -0.68), with similar findings at the hospital influent level (Spearman's rho = -0.61). These results suggest that wastewater surveillance at public tertiary care hospitals may provide valuable complementary data to municipal WWTPs, particularly in LMICs where population coverage of municipal WWTPs is low.

Integrating WBE into hospital infectious disease sentinel surveillance systems can enhance clinical monitoring while reducing costs. During the study period, the cost for

RT-PCR testing was $40 per individual clinical sample, compared to $45 for each pooled wastewater sample from multiple patients. This highlights the economic benefits of using wastewater data for complimentary surveillance alongside clinical testing, especially in LMIC settings.

The hospital campus was selected for investigation due to its strong potential to serve as a long-term sentinel surveillance site for Bangkok, Thailand. With over 1,500 inpatient beds and approximately 1.5 million outpatients served annually, its centrally located position and large patient capacity allow for a comprehensive representation of clinical disease within one of Southeast Asia's largest cities. Clinical COVID-19 case and lineage data from the selected hospital closely mirrored national trends (Figs 2 and 3). Further investigation is needed to better understand the demographics of individuals contributing to wastewater in public tertiary hospitals in Thailand. This information is crucial to determine whether hospital wastewater captures populations excluded in municipal WWTP surveillance.

With a diverse array of human pathogens detected and quantified in wastewater, expanding wastewater surveillance in hospital settings beyond SARS-CoV-2 is becoming increasingly feasible [40]. A Chicago-based study conducted between March 13 and June 26, 2023, consistently detected mpox in wastewater before reported cases, including at healthcare facilities such as congregate living facilities [41]. When selecting sample sites specific to certain clinical use types, it is important to account for variations in patient demographics that influence pathogen contributions captured in wastewater. This highlights the importance of further clinical and wastewater investigations into the detection and quantification of pathogen contributions from varying shedding pathways and across diverse populations, especially among complex and severe infections likely present in a hospital setting.

Reliable SARS-CoV-2 detection in wastewater samples was ensured through stringent quality control measures in this study, including positive and negative controls, and an internal control to verify RNA extraction efficiency. Furthermore, the sampling, transportation, and storage procedures were carefully managed to preserve sample integrity. These comprehensive measures contributed to the consistency and validity of the results reported throughout the investigation.

This study faces several limitations. First, we investigated only one hospital campus, which may not fully represent the diverse resource levels, infrastructure, and patient demographics of other public tertiary hospitals in the region. To better understand the generalizability of our conclusions, further research should include multiple hospital settings, particularly major tertiary care hospitals (1,000+ beds) across Thailand. Second, our sampling schedule restricted us to once-per-week collection, limiting our ability to assess the benefits of more frequent sampling. Although our wastewater data indicated similar detection trends across sample sites at this frequency, this pattern did not persist during periods of low reported COVID-19 cases (February to early March 2023). This suggests that more frequent sampling may be beneficial for capturing trends when a disease is rare or emerging. Third, due to the dynamic nature of pandemic-related policies at the selected hospital, such as changes in building usage and testing protocols, there was a lack of consistent records available for these variables. As a result, controlling for these shifts was challenging and may introduce potential bias in our analysis. Finally, the absence of nosocomial COVID-19 outbreak data within the selected hospital's online database limited our ability to definitively link wastewater spikes to hospital-acquired infections.

## Conclusions

Our 47-week observational study establishes the feasibility of multi-scale WBE for SARS-CoV-2 surveillance in a public tertiary care hospital in Bangkok. We found significant

correlations between clinical COVID-19 case data and wastewater N gene Ct values across various sampling locations, including buildings, building clusters, and the hospital campus. Our results hold promise for designing cost-effective wastewater surveillance programs in large tertiary care hospitals. Influent wastewater at the hospital-campus level emerged as the most consistent indicator of SARS-CoV-2 presence highlighting its potential for early detection of novel or emerging pathogens. For more granular investigations within the hospital, targeting wastewater from hospital buildings may provide valuable data related to isolated outbreaks or viral shedding patterns among specific patient and non-patient populations. Furthermore, our findings suggest that hospital wastewater analysis can provide valuable insights into disease prevalence beyond the hospital itself, especially in LMIC settings where a significant portion of wastewater bypasses municipal WWTPs.

## Supporting information

**S1 Table. Hospital Twice Monthly COVID-19 Patient Case Report.**
(DOCX)

**S2 Table. Hospital Weekly COVID-19 Personnel Case Report.**
(DOCX)

**S3 Table. WHO Thailand Weekly COVID-19 Case Report.**
(DOCX)

**S4 Table. GISAID Accession Numbers of SARS-CoV-2 Genomic Surveillance Data in the Selected Hospital.**
(DOCX)

**S5 Table. Raw RT-PCR Wastewater Data.**
(DOCX)

**S6 Table. Spearman's Rho Coefficient for the Correlation Between Wastewater Sample SARS-CoV-2 N Gene Ct Values at Various Sample Sites and COVID-19 Clinical Reporting at the Hospital and National Level. An asterisk indicates Z-test statistical significance after Bonferroni correction.**
(DOCX)

## Acknowledgments

We would like to express our deep appreciation to Ms. Sininat Petcharat for preparing the project and technical support. We would also like to thank the personnel from the Facilities Department, King Chulalongkorn Hospital for their help with sample collection.

## Author contributions

**Conceptualization:** Quinton Hayre, Opass Putcharoen, Leilani Paitoonpong.

**Data curation:** Quinton Hayre, Anaporn Supataragul.

**Formal analysis:** Quinton Hayre.

**Funding acquisition:** Supaporn Wacharapluesadee, Opass Putcharoen, Leilani Paitoonpong.

**Investigation:** Piyapha Hirunpatrawong.

**Methodology:** Supaporn Wacharapluesadee, Leilani Paitoonpong.

**Project administration:** Supaporn Wacharapluesadee, Leilani Paitoonpong.

**Resources:** Supaporn Wacharapluesadee.

**Supervision:** Opass Putcharoen.

**Visualization:** Quinton Hayre.

**Writing – original draft:** Quinton Hayre.

**Writing – review & editing:** Quinton Hayre, Supaporn Wacharapluesadee, Opass Putcharoen, Leilani Paitoonpong.

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
