## [Decision Letter · Decision Letter 0]

17 Dec 2024

PGPH-D-24-02707

Multi-scale wastewater surveillance at a Bangkok tertiary care hospital: A potential sentinel site for real-time COVID-19 surveillance at hospital and national levels

Dear Dr. Leilani Paitoonpong

Thank you for submitting your manuscript to PLOS Global Public Health. After careful consideration, we feel that it has merit but does not fully meet PLOS Global Public Health’s publication criteria as it currently stands. Therefore, we invite you to submit a revised version of the manuscript that addresses the points raised during the review process.

We look forward to receiving your revised manuscript.

Kind regards,

Muhammad Asaduzzaman, MD MPH MPhil

Academic Editor

Additional Editor Comments (if provided):

Reviewers' comments:

Reviewer's Responses to Questions

**Comments to the Author**

1. Does this manuscript meet PLOS Global Public Health’s publication criteria ? Is the manuscript technically sound, and do the data support the conclusions? The manuscript must describe methodologically and ethically rigorous research with conclusions that are appropriately drawn based on the data presented.

Reviewer #1: Partly

Reviewer #2: Yes

Reviewer #3: Partly

2. Has the statistical analysis been performed appropriately and rigorously?

Reviewer #1: Yes

Reviewer #2: Yes

Reviewer #3: Yes

3. Have the authors made all data underlying the findings in their manuscript fully available (please refer to the Data Availability Statement at the start of the manuscript PDF file)?

Reviewer #1: Yes

Reviewer #2: Yes

Reviewer #3: Yes

4. Is the manuscript presented in an intelligible fashion and written in standard English?

Reviewer #1: Yes

Reviewer #2: Yes

Reviewer #3: No

5. Review Comments to the Author

Reviewer #1: Dear Editor,

I have reviewed the manuscript "Multi-scale wastewater surveillance at a Bangkok tertiary care hospital: A potential sentinel site for real-time COVID-19 surveillance at hospital and national levels" and provide my detailed assessment below.

OVERALL ASSESSMENT

This manuscript presents a valuable contribution to wastewater-based epidemiology, particularly for healthcare settings in regions with limited municipal wastewater infrastructure. The study design is methodologically sound, and the findings have important implications for public health surveillance. While the paper requires some revisions, the core research and conclusions are scientifically valid and worthy of publication.

MAJOR STRENGTHS

1. Novel multi-scale approach examining wastewater surveillance at different spatial resolutions

2. Comprehensive sampling strategy (n=392) over 47 weeks

3. Strong correlation analysis between wastewater data and clinical cases

4. Clear relevance for low- and middle-income countries

5. Well-structured presentation of findings

MAJOR CONCERNS

1. Methodological Details

- RNA extraction efficiency and quality control measures need elaboration

- PCR inhibition controls should be described

- Sample storage/transport conditions require more detail

- Building occupancy data and its potential impact on results should be discussed

2. Statistical Analysis

- Multiple testing corrections should be applied

- Time-series analysis would strengthen temporal correlations

- Power analysis should be included

- Confidence intervals for correlation coefficients needed

- Outlier analysis and sensitivity testing warranted

3. Data Availability

- Raw RT-qPCR data should be provided

- Quality control metrics need inclusion

- Building-specific metadata would be valuable

- Database accession numbers for genomic data required

SPECIFIC RECOMMENDATIONS

Methods Section:

1. Add details about:

- RNA extraction controls and efficiency measurements

- PCR inhibition testing protocols

- Sample storage conditions and temperature monitoring

- Transport time between collection and processing

2. Include:

- Building occupancy data if available

- Wastewater flow rate measurements

- Temperature and pH measurements of samples

- Detailed quality control procedures

Results Section:

1. Strengthen by adding:

- Multiple testing corrections for correlations

- Confidence intervals for all statistical measures

- Formal outlier analysis

- Sensitivity analyses

- Time-series analysis of temporal trends

2. Expand discussion of:

- Potential confounding factors

- Seasonal effects on viral detection

- Impact of building usage patterns

- Changes in clinical testing practices

Discussion Section:

1. Address:

- Generalizability to other hospital settings

- Cost-effectiveness compared to clinical surveillance

- Limitations of the approach

- Practical implementation considerations

2. Include:

- Recommendations for sampling frequency

- Guidelines for site selection

- Quality control requirements

- Resource implications

- Add new figure showing temporal trends with confidence intervals

MINOR ISSUES

1. Editorial:

- Some redundancy in methods section

- Inconsistent abbreviation use

- Variable formatting of units

- References need updating

2. Clarity:

- Better define building categories

- Explain sampling time selection

- Clarify quality control criteria

- Expand on statistical methods

RECOMMENDATIONS FOR AUTHORS

1. Provide complete methodological details

2. Strengthen statistical analysis

3. Include all raw data and quality control metrics

4. Expand discussion of limitations

5. Address practical implementation considerations

PUBLICATION RECOMMENDATION

Accept with Major Revisions

The manuscript presents valuable research with important implications for public health surveillance. The suggested revisions would strengthen the paper's scientific rigor and utility to the field. The authors should:

1. Address all methodological concerns

2. Strengthen statistical analysis

3. Provide complete data

4. Expand discussion of limitations and practical implementation

With these revisions, this paper would make a contribution to the literature on wastewater-based epidemiology and would be suitable for publication in PLOS Global Public Health.

Reviewer #2: The research on multi-scale waste water surveillance at a Bangkok tertiary hospital added another insight into the relevance of environmental surveillance as another pathway to early detection of some pathogenic organisms such as SARS Cov-2 in waste water and this could improve onsurveillance of other water-borne diseases. The study was well thought out and well implemented. The language was scientific and easy to understand. The use of statistical packages to support findings is okay. The conclusion was direct at the objectives.

Reviewer #3: The abstract needs to be written in an organized and unambiguous manner.

113: “Central location” should be changed to centrally located

134: There is fig. 1 without the provision of such figure

The study design is not cleared

Ethical approval was not obtained from the IRB of the hospital before the collection of clinical data

6. PLOS authors have the option to publish the peer review history of their article (what does this mean? ). If published, this will include your full peer review and any attached files.

**Do you want your identity to be public for this peer review?** For information about this choice, including consent withdrawal, please see our Privacy Policy .

Reviewer #1: No

Reviewer #2: No

Reviewer #3: No

---

## [Decision Letter · Decision Letter 1]

27 Feb 2025

Multi-scale wastewater surveillance at a Bangkok tertiary care hospital: A potential sentinel site for real-time COVID-19 surveillance at hospital and national levels

PGPH-D-24-02707R1

Dear Leilani Paitoonpong,

We are pleased to inform you that your manuscript 'Multi-scale wastewater surveillance at a Bangkok tertiary care hospital: A potential sentinel site for real-time COVID-19 surveillance at hospital and national levels' has been provisionally accepted for publication in PLOS Global Public Health.

Best regards,

Muhammad Asaduzzaman, MD MPH MPhil

Academic Editor

Reviewer Comments (if any, and for reference):

Reviewer's Responses to Questions

**Comments to the Author**

1. If the authors have adequately addressed your comments raised in a previous round of review and you feel that this manuscript is now acceptable for publication, you may indicate that here to bypass the “Comments to the Author” section, enter your conflict of interest statement in the “Confidential to Editor” section, and submit your "Accept" recommendation.

Reviewer #1: All comments have been addressed

Reviewer #3: All comments have been addressed

2. Does this manuscript meet PLOS Global Public Health’s publication criteria ? Is the manuscript technically sound, and do the data support the conclusions? The manuscript must describe methodologically and ethically rigorous research with conclusions that are appropriately drawn based on the data presented.

Reviewer #1: Yes

Reviewer #3: Yes

3. Has the statistical analysis been performed appropriately and rigorously?

Reviewer #1: Yes

Reviewer #3: Yes

4. Have the authors made all data underlying the findings in their manuscript fully available (please refer to the Data Availability Statement at the start of the manuscript PDF file)?

Reviewer #1: Yes

Reviewer #3: Yes

5. Is the manuscript presented in an intelligible fashion and written in standard English?

Reviewer #1: Yes

Reviewer #3: Yes

6. Review Comments to the Author

Reviewer #1: Overall, the manuscript presents a valuable contribution to the field of wastewater-based epidemiology for public health surveillance. The authors have clearly described a robust multi-scale sampling strategy that captures SARS-CoV-2 RNA levels from various sites within a large tertiary care hospital, and have successfully correlated these findings with clinical case data at both the hospital and national levels.

**Methodological and Technical Soundness:**

- The study’s design is innovative, using sampling at building, cluster, and campus levels to gain a comprehensive picture of viral dynamics.

- The methodology is detailed and rigorous. The authors employ standardized RT-PCR techniques with appropriate quality control measures, ensuring reliable detection of SARS-CoV-2 RNA.

- The statistical analysis is conducted with care—using Spearman’s rank correlation, Bonferroni corrections for multiple comparisons, and Fisher Z-transformations for confidence intervals—thereby supporting the conclusions with strong evidence.

**Data Transparency and Availability:**

- In compliance with PLOS Global Public Health’s data policy, all data underlying the study’s findings are made fully available. Wastewater data are included in the manuscript and supplementary materials, and additional clinical data are accessible via the WHO dashboard. This level of transparency is commendable and enhances the reproducibility of the research.

**Presentation and Clarity:**

- The manuscript is written in clear, standard English with an intelligible structure that facilitates comprehension. Minor typographical or grammatical errors are present but do not detract from the overall clarity of the work.

**Additional Considerations:**

- The study acknowledges some limitations, such as the once-weekly sampling schedule and potential gaps in capturing contributions from all relevant population groups (e.g., non-reporting hospital staff). While these factors may slightly limit the granularity of temporal trends, they are appropriately discussed and do not compromise the overall validity of the conclusions.

- There are no concerns regarding dual publication, research ethics, or publication ethics. The authors have addressed previous reviewer comments adequately, and the manuscript now meets the ethical and methodological standards required for publication.

In summary, the manuscript is methodologically rigorous, the data robustly support the conclusions, and the study is clearly presented. I recommend acceptance for publication after minor revisions addressing the minor issues noted above.

Reviewer #3: Significant changes have been made to the abstract.

This includes organizing my thoughts sequentially from background to methods to results to the

study’s conclusions. Additionally, further detail has been added to provide less ambiguity

7. PLOS authors have the option to publish the peer review history of their article (what does this mean? ). If published, this will include your full peer review and any attached files.

**Do you want your identity to be public for this peer review?** For information about this choice, including consent withdrawal, please see our Privacy Policy .

Reviewer #1: No

Reviewer #3: **Yes: ** P.D.Adewole
